# Trick or Treat(ment): Should We Still Fear Reperfusion Therapy in Anticoagulated Stroke Patients?—Comparable 90-Day Outcomes in a Propensity-Score-Matched Registry Study

**DOI:** 10.3390/jcm14228146

**Published:** 2025-11-17

**Authors:** Jessica Seetge, Balázs Cséke, Zsófia Nozomi Karádi, Edit Bosnyák, Eszter Johanna Jozifek, László Szapáry

**Affiliations:** 1Stroke Unit, Department of Neurology, University of Pécs, Ifjúság útja 13, 7624 Pécs, Baranya, Hungary; karadi.zsofia@pte.hu (Z.N.K.); bosnyak.edit@pte.hu (E.B.); jozifek.eszter@pte.hu (E.J.J.); 2Department of Emergency Medicine, University of Pécs, Ifjúság útja 13, 7624 Pécs, Baranya, Hungary; cseke.balazs@pte.hu

**Keywords:** acute ischemic stroke, anticoagulated patients, functional outcomes, reperfusion therapy, propensity score matching

## Abstract

**Background:** The management of acute ischemic stroke (AIS) in anticoagulated patients presents a clinical challenge, as concerns about safety and efficacy often limit access to recanalization therapies. Despite the widespread use of direct oral anticoagulants (DOACs) and vitamin K antagonists (VKAs), their impact on functional recovery and mortality following intravenous thrombolysis (IVT) and mechanical thrombectomy (MT) remains uncertain. Therefore, this study investigates the association between prior anticoagulation and 90-day outcomes in AIS patients undergoing reperfusion therapy. **Methods:** We conducted a retrospective cohort analysis using our institutional stroke registry, including AIS patients admitted to the Department of Neurology at our university between February 2023 and 2025. Anticoagulated patients were 1:1 propensity score-matched with non-anticoagulated controls (*n* = 126 per group) using Mahalanobis distance matching with a caliper, adjusting for age, sex, hypertension, diabetes, stroke severity (National Institutes of Health Stroke Scale [NIHSS] at admission and 72 h), and pre-stroke functional status (pre-morbid modified Rankin Scale [pre-mRS]). Primary endpoints at 90 days were functional independence (modified Rankin Scale [mRS] ≤ 2), mRS-shift, and mortality (mRS = 6). Predictors of outcome were assessed using multivariable logistic regression and generalized additive models (GAMs). Subgroup analyses evaluated the effects of anticoagulation type and treatment modality. **Results:** Among 866 AIS patients (DOAC *n* = 100, VKA *n* = 48, non-anticoagulated *n* = 718), 426 (49.2%) underwent reperfusion therapy (IVT *n* = 195, MT *n* = 163, IVT + MT *n* = 68). Before matching, anticoagulated patients were less likely to achieve functional independence (34.5% vs. 52.1%, odds ratio [OR] = 0.48, 95% confidence interval [CI] [0.33–0.70], *p* < 0.001), had a greater mRS-shift (2.53 vs. 1.79, *p* < 0.001), and higher mortality (30.4% vs. 14.5%, OR = 2.58, 95% CI [1.72–3.88], *p* < 0.001). However, after matching, these differences were no longer statistically significant. NIHSS, 72hNIHSS, and pre-mRS were the strongest independent predictors of outcome (*p* < 0.001), while anticoagulation status had no significant effect. **Conclusions:** Recanalization therapy was not associated with worse functional outcomes in selected anticoagulated AIS patients. These findings suggest that prior anticoagulation alone should not preclude reperfusion therapy in otherwise eligible patients, and underscore the importance of individualized, evidence-based decision-making in acute stroke care.

## 1. Introduction

Acute ischemic stroke (AIS) remains one of the leading causes of death and disability worldwide [1]. Time-sensitive reperfusion therapies, such as intravenous thrombolysis (IVT) and mechanical thrombectomy (MT), are the cornerstone of acute stroke management, offering significantly improved outcomes when administered promptly [2,3]. Oral anticoagulants (OACs), including vitamin K antagonists (VKAs) and direct oral anticoagulants (DOACs), effectively reduce the risk of cardioembolic stroke, particularly in patients with atrial fibrillation [4]. However, their presence during the acute phase introduces clinical challenges, as concerns over an increased risk of intracranial hemorrhage (ICH) complicate decisions regarding reperfusion therapy [5,6]. Consequently, clinicians are often faced with considerable uncertainty when determining the safest and most effective treatment strategy for anticoagulated stroke patients.

Current guidelines, including those from the American Heart Association/American Stroke Association (AHA/ASA) and the European Stroke Organisation (ESO), outline clear eligibility criteria for administering IVT in anticoagulated patients. Specifically, IVT is considered safe in VKA users with an international normalized ratio (INR) ≤ 1.7, and in DOAC-treated patients if at least 48 h have passed since the last dose (assuming normal renal function) or if specific coagulation assays (e.g., anti-Xa activity, thrombin time) indicate adequate anticoagulant clearance [7,8]. MT, by contrast, has no formal contraindications related to anticoagulation status [3,7].

Despite these permissive recommendations, real-world practice continues to reflect substantial hesitancy in offering reperfusion therapy to anticoagulated patients, even when they meet eligibility criteria. Among AIS patients on DOACs, an estimated 28% are eligible for IVT (presenting within 4.5 h and with a National Institutes of Health Stroke Scale [NIHSS] score ≥ 4) [9,10]. However, registry data from Germany [11] and Switzerland [12] show that only 6–15% of eligible patients actually receive IVT, with anticoagulation frequently cited as the main reason for withholding treatment.

While much of the debate has focused on expanding access to reperfusion beyond current guideline-defined thresholds, a more immediate and underexplored question remains: Do anticoagulated patients who meet all existing eligibility criteria for reperfusion therapy actually have worse outcomes than their non-anticoagulated counterparts? If the answer is no, then withholding treatment solely based on anticoagulation status, even in otherwise eligible individuals, may reflect an overly cautious approach not justified by current evidence.

To address this critical question, we conducted a retrospective analysis using a large, single-center stroke registry. Employing rigorous propensity score matching, we compared 90-day outcomes, including functional independence, mRS-shift, and mortality, between anticoagulated and non-anticoagulated AIS patients who received reperfusion therapy in accordance with current guideline recommendations. Our aim was to determine whether anticoagulation status independently affects clinical outcomes and to evaluate whether therapeutic hesitation in this population is truly supported by data.

## 2. Materials and Methods

### 2.1. Study Design

A retrospective cohort analysis was conducted using the TINL (Transzlációs Idegtudományi Nemzeti Laboratórium) STROKE-registry, which included patients admitted with AIS to the Department of Neurology at the University of Pécs between February 2023 and 2025.

### 2.2. Data Collection and Measurements

Baseline characteristics included demographic data (age, sex), clinical variables such as pre-stroke functional status (pre-morbidity modified Rankin Scale [pre-mRS] scores), and stroke severity (assessed by the NIHSS at both admission and 72 h post-stroke) [13], ICH status, stroke etiology, onset-to-door-time, and plasma glucose levels. Comorbidities, including hypertension and diabetes mellitus, were documented. In anticoagulated patients, additional information was collected regarding the type of anticoagulant (DOAC vs. VKA), presence of atrial fibrillation, and history of prior stroke. Treatment modalities (IVT, MT or combined therapy [IVT + MT]) were also recorded.

### 2.3. Inclusion and Exclusion Criteria

A total of 1102 patients with AIS were initially assessed. Of these, 236 were excluded due to incomplete data: 233 had missing 90-day modified Rankin Scale (mRS) scores, 2 lacked admission NIHSS scores, and 1 had no recorded 72-h NIHSS score. The final study population consisted of 866 patients with complete baseline and outcome data.

### 2.4. Caliper-Matched Propensity Score Matching

To minimize baseline differences and confounding between anticoagulated and non-anticoagulated patients, 1:1 propensity score matching was performed using Mahalanobis distance within a caliper. Matching variables included age, sex, hypertension, diabetes, pre-mRS, and NIHSS scores at admission and at 72 h. An initial 148 matched pairs (296 patients) were identified, of which 126 high-quality pairs (252 patients) were retained after caliper restriction. Balance between matched groups was assessed using standardized mean differences (SMDs), variance ratios (VRs), and the Kolmogorov–Smirnov test.

### 2.5. Outcome Assessment

The primary outcomes were functional independence at 90 days, defined as an mRS score of ≤2, overall functional trajectory, assessed as mRS-shift (i.e., the change from pre-mRS to 90-day mRS); and 90-day mortality, defined as an mRS score of 6 [14]. Secondary analyses explored whether outcomes varied by type of anticoagulation (DOAC vs. VKA) or by reperfusion strategy (IVT, MT, or IVT + MT).

### 2.6. Statistical Analysis

All statistical analyses were performed using Python (version 3.14.0). Continuous variables were assessed for normality using the Shapiro–Wilk test and reported as either mean ± standard deviation (SD) for normally distributed data or median with interquartile range (IQR) for non-normally distributed data. Categorical variables were summarized as absolute frequencies and percentages. Baseline characteristics between anticoagulated and non-anticoagulated patients were compared using Fisher’s exact test or Chi-square test for categorical variables. For continuous variables, independent samples *t*-tests were used when normally distributed, and the Kruskal–Wallis test was applied for non-normally distributed variables.

Clinical outcomes were analyzed both before and after matching. For binary outcomes, including functional independence at 90 days and mortality, group comparisons were conducted using Chi-square or Fisher’s exact tests, as appropriate. Corresponding odds ratios (ORs) and 95% confidence intervals (CIs) were calculated to assess effect sizes. Ordinal analysis of mRS-shift was performed using the Mann–Whitney U test, and treatment group comparisons were evaluated using the Kruskal–Wallis test.

To identify predictors of outcomes, multivariable logistic regression was applied to the matched cohort for both functional independence and mortality. Functional trajectory, assessed as mRS-shift across the full range of the scale, was analyzed using generalized additive models (GAMs) in the matched cohort.

Within the anticoagulated subgroup, separate analyses were performed to identify predictors of favorable outcome, mRS-shift, and mortality using the same modeling approaches. Sensitivity analyses included variance inflation factor (VIF) assessment to detect multicollinearity, and E-value calculations to estimate the potential influence of unmeasured confounders on observed associations.

### 2.7. Ethics Approval

This study was conducted in accordance with the Declaration of Helsinki and reviewed and approved by the Scientific and Research Ethics Committee of the Medical Research Council of the University of Pécs (RRF-2.3.1-21-2022-00011, 1 September 2022) and the Scientific and Research Ethics Committee of the Medical Research Council of Hungary (BM/22444-1/2024, 1 September 2024) to ensure that it met all regulatory requirements and ethical guidelines, including participant privacy and data protection standards. All study procedures were carried out in compliance with applicable ethical guidelines, and ongoing monitoring by the ethics committees ensured adherence to approved protocols.

## 3. Results

The final cohort included 866 patients with acute ischemic stroke, of whom 148 (17.1%) were receiving oral anticoagulation prior to admission, 100 (11.5%) DOACs and 48 (5.5%) VKAs. The remaining 718 patients (82.9%) had no history of anticoagulant use. Reperfusion therapy was administered to 426 patients (49.2%), including 195 who received IVT, 163 treated with MT, and 68 who underwent both IVT and MT.

### 3.1. Baseline and Clinical Characteristics

Before matching, anticoagulated patients were significantly older than non-anticoagulated patients (mean age: 76.5 ± 11.1 vs. 69.7 ± 12.1 years, *p* < 0.001). They also had worse pre-stroke functional status (median pre-mRS: 0 [0–2] vs. 0 [0–1], *p* = 0.018) and presented with more severe strokes at 72 h (median NIHSS: 4 [1–11] vs. 2 [0–7], *p* = 0.030). Cardioembolic strokes were significantly more common among anticoagulated patients (73.7% vs. 26.9%, *p* < 0.001), as were hypertension (93.9% vs. 80.8%, *p* < 0.001) and diabetes mellitus (39.9% vs. 34.0%, *p* = 0.020).

Differences in recanalization therapy were also evident: anticoagulated patients were much less likely to receive IVT (3.4% vs. 26.5%, *p* < 0.001), but more likely to undergo MT (26.4% vs. 17.3%, *p* = 0.014). Rates of combined IVT and MT did not significantly differ between groups (*p* = 0.086). A summary of these baseline differences is presented in Table 1.

After matching, most demographic and clinical characteristics were well balanced between groups. All key covariates demonstrated SMDs below 0.1, indicating minimal residual imbalance (Table 2). In addition, VRs were close to 1.0, and Kolmogorov–Smirnov test *p*-values showed no significant distributional differences between groups (Table 3).

However, a few significant differences persisted after matching. Cardioembolic stroke etiology remained more prevalent in the anticoagulated group (75.4% vs. 30.2%, *p* < 0.001), IVT was still administered less frequently (4.0% vs. 30.9%, *p* < 0.001), and MT was more commonly performed (25.4% vs. 12.7%, *p* = 0.016). Crucially, subsequent multivariable regression and GAM analyses demonstrated that neither cardioembolic stroke etiology nor treatment modality was independently associated with favorable outcomes, mRS-shift, or mortality, suggesting that these residual imbalances did not confound the study’s primary findings.

### 3.2. Favorable Outcome

Before matching, anticoagulated patients were significantly less likely to achieve functional independence at 90 days, with only 34.5% (95% CI [27.3–42.4%]) reaching an mRS ≤ 2, compared to 52.1% (95% CI [48.4–55.7%]) in the non-anticoagulated group (*p* < 0.001). The corresponding OR = 0.48 (95% CI [0.33–0.70], *p* < 0.001) indicated that anticoagulated patients had 52% lower odds of a favorable outcome.

After matching, this difference was no longer statistically significant. In the matched cohort, 39.7% (95% CI [31.6–48.4%]) of anticoagulated patients achieved an mRS ≤ 2, compared to 45.2% (95% CI [36.8–53.9%]) in non-anticoagulated patients (*p* = 0.445). The OR = 0.80 (95% CI [0.48–1.31]) suggested no independent association between anticoagulation and functional outcome.

For the comparison between DOAC- and VKA-treated patients, no significant differences were observed. Before matching, the OR = 0.82 (95% CI [0.40–1.68], *p* = 0.586), and after matching, the OR remained similar (0.82, 95% CI [0.39–1.74], *p* = 0.700), indicating that anticoagulant type did not influence 90-day outcomes.

Regarding treatment modalities, prior to matching, IVT was associated with higher odds of functional independence compared to SC (OR = 2.66, 95% CI [1.86–3.81], *p* < 0.001), while MT was linked to lower odds (OR = 0.47, 95% CI [0.32–0.69], *p* < 0.001). Combination therapy showed no significant benefit (OR = 1.06, 95% CI [0.64–1.77], *p* = 0.896). After matching, these associations were attenuated: IVT (OR = 1.48, 95% CI [0.75–2.91], *p* = 0.302) and IVT + MT (OR = 0.62, 95% CI [0.20–1.89], *p* = 0.428). Although MT remained significant in univariate analysis (OR = 0.46, 95% CI [0.22–0.94], *p* = 0.041), the association did not persist in multivariable regression.

When adjusting for covariates in the matched cohort, including variables not fully balanced by matching such as stroke etiology and treatment modality, the adjusted OR (aOR = 0.64, 95% CI [0.26–1.61], *p* = 0.346) confirmed the absence of a significant association between anticoagulation status and functional independence.

### 3.3. mRS-Shift

The Shapiro–Wilk test confirmed non-normal distribution of mRS-shift scores in both groups (*p* < 0.001), supporting the use of non-parametric methods.

Before matching, anticoagulated patients had a significantly greater mean mRS-shift (2.53 ± 2.23) compared to non-anticoagulated patients (1.79 ± 1.96, *p* < 0.001), indicating more pronounced functional decline. After matching, this difference was attenuated and no longer statistically significant (2.31 ± 2.25 vs. 1.87 ± 1.91, *p* = 0.227).

In the anticoagulated subgroup, DOAC users had numerically higher mRS-shift than VKA users, though differences were not statistically significant either before (2.76 ± 2.22 vs. 2.04 ± 2.18, *p* = 0.060) or after matching (2.54 ± 2.25 vs. 1.86 ± 2.19, *p* = 0.086).

Among treatment modalities, IVT was associated with a lower mRS-shift compared to SC before matching (1.21 ± 1.62 vs. 1.88 ± 2.02, *p* < 0.001), but this difference was no longer significant after matching (*p* = 0.630). MT was linked to significantly greater mRS-shift both before (2.74 ± 2.07 vs. 1.88 ± 2.02, *p* < 0.001) and after matching (2.98 ± 2.06 vs. 1.88 ± 2.12, *p* = 0.002). Combination therapy showed a significant increase only after matching (3.20 ± 2.37 vs. 1.88 ± 2.12, *p* = 0.041), though not before (*p* = 0.252). However, none of these associations remained significant in multivariable analyses.

After adjusting for covariates in the matched sample, including variables not fully balanced through matching, such as cardioembolic etiology and recanalization therapy, no significant difference in mRS-shift was observed between anticoagulated and non-anticoagulated patients (adjusted coefficient = 0.29, 95% CI [−0.18–0.76], *p* = 0.223).

### 3.4. Mortality

Before matching, mortality was significantly higher among anticoagulated patients (30.4%) compared to non-anticoagulated patients (14.5%), with OR = 2.58 (95% CI [1.72–3.88], *p* < 0.001). After matching, mortality remained elevated (22.2% vs. 11.9%), with a still significant OR = 2.11 (95% CI [1.07–4.19], *p* = 0.044), though the effect size was reduced.

In the anticoagulated subgroup, no significant differences in mortality were found between DOAC and VKA users. Pre-matching OR = 1.27 (95% CI [0.59–2.72], *p* = 0.573), and post-matching OR = 1.33 (95% CI [0.53–3.33], *p* = 0.652), indicating no meaningful variation by anticoagulant type.

Patients receiving IVT had significantly lower mortality than those receiving SC before matching (7.2% vs. 19.1%, OR = 0.33, 95% CI [0.18–0.59], *p* < 0.001). While this trend persisted after matching (6.8% vs. 17.9%), it no longer reached statistical significance (OR = 0.33, 95% CI [0.10–1.17], *p* = 0.094). MT was associated with slightly higher mortality than SC both before (OR = 1.33, 95% CI [0.87–2.05], *p* = 0.211) and after matching (OR = 1.20, 95% CI [0.53–2.72], *p* = 0.672), but neither result was significant. Combination therapy also showed no mortality benefit, with OR = 0.91 (95% CI [0.47–1.77], *p* = 0.869) before and OR = 1.66 (95% CI [0.49–5.64], *p* = 0.485) after matching.

When adjusting for covariates in the matched cohort, including stroke etiology and treatment modality, the association between anticoagulation and mortality was no longer statistically significant (aOR = 2.35, 95% CI [0.79–7.02], *p* = 0.125).

### 3.5. Predictors of Outcome in the Matched Cohort

#### 3.5.1. Favorable Outcome

Multivariable logistic regression analysis (Table 4) identified NIHSS at admission (*p* = 0.035), NIHSS at 72 h (*p* < 0.001), pre-mRS (*p* < 0.001), and sex (*p* = 0.043) as significant predictors of favorable functional outcome. Specifically, being male was associated with higher odds of favorable outcomes compared to females. Anticoagulation status was not a significant predictor (*p* = 0.346). The model demonstrated a good fit (Pseudo R^2^ = 0.4949) and a highly significant overall model effect (Log-Likelihood Ratio [LLR], *p* = 4.691e-30), supporting the robustness of these predictors.

In terms of predictive performance (Figure 1), the model showed excellent discrimination with an area under the curve (AUC) of 0.93, an accuracy of 86.5%, sensitivity of 85.5%, and specificity of 87.9%. Precision was high at 90.5%, with an F1 score of 87.9%, indicating strong reliability in identifying patients at risk of poor functional recovery. Calibration was also good, reflected by a Brier Score of 0.1055. The optimal classification threshold, determined via Youden’s Index, was 0.50.

#### 3.5.2. mRS-Shift

Higher NIHSS scores at admission (*p* = 0.026), NIHSS at 72 h (*p* < 0.001), and pre-stroke mRS (*p* < 0.001) were significant predictors of increased mRS-shift, indicating worse functional outcomes. Anticoagulation status was not significantly associated with functional outcomes (*p* = 1.00).

The GAM demonstrated strong explanatory power, as indicated by a high Pseudo R^2^ of 0.612, effective degrees of freedom (DoF) of 33.61, and a log-likelihood of −456.25. An Akaike Information Criterion (AIC) of 981.71 further supported the model’s optimal balance between complexity and fit.

GAM-derived plots (Figure 2) highlighted non-linear associations between continuous predictors and mRS-shifts. Elevated NIHSS scores at admission and at 72 h consistently predicted greater mRS-shift.

#### 3.5.3. Mortality

Logistic regression analysis (Table 5) identified NIHSS scores at admission and at 72 h (both *p* < 0.001) as significant independent predictors of mortality. In contrast, anticoagulation status was not significantly associated with mortality risk (*p* = 0.125). The model exhibited strong overall performance, with a Pseudo R^2^ of 0.3860, a log-likelihood of −70.69 (compared to LL-Null of −115.14), and a highly significant likelihood ratio test (LLR *p* = 8.083 × 10^−14^), indicating robust model fit.

Regarding discrimination (Figure 3), the model achieved an excellent AUC of 0.92, with an accuracy of 80%, sensitivity of 98%, and specificity of 76%. Precision was 0.46, and the F1 score was 0.62, highlighting the model’s strong capability in predicting mortality while managing false positives effectively. Calibration was also robust, as demonstrated by a low Brier Score of 0.0905. The optimal classification threshold, based on Youden’s Index, was determined to be 0.11.

### 3.6. Predictors of Outcome in Anticoagulated Patients

#### 3.6.1. Favorable Outcome

Within the anticoagulated cohort, multivariable logistic regression analysis (Table 6) identified NIHSS scores at 72 h (*p* < 0.001) and pre-stroke mRS (*p* = 0.001) as the strongest predictors of favorable functional outcomes. Neither treatment modality nor the type of anticoagulation significantly influenced the risk of favorable recovery. Atrial fibrillation and previous stroke were included due to their established clinical relevance in anticoagulated patients, although neither was a significant predictor in this model. The model demonstrated excellent fit and robustness, with a high Pseudo R^2^ of 0.5815, a log-likelihood of −35.42 (compared to LL-Null of −84.64), and a highly significant likelihood ratio test (LLR *p* = 9.461 × 10^−15^).

In terms of discriminative performance, the model exhibited strong predictive capabilities, achieving an AUC of 0.95, accuracy of 90%, sensitivity of 92%, specificity of 88%, precision of 84%, and an F1 score of 0.88. These metrics underscore the model’s reliability in distinguishing patients who experienced unfavorable outcomes from those who did not. Calibration was also excellent, as indicated by a Brier Score of 0.0846.

#### 3.6.2. mRS-Shift

To further explore predictors of disability progression within the anticoagulated cohort, GAMs were applied to assess mRS-shift. NIHSS at 72 h (*p* < 0.001) and pre-stroke mRS (*p* = 0.001) emerged as significant predictors of worsening functional status. In contrast, treatment modality and type of anticoagulation were not significantly associated with mRS-shift. The model demonstrated strong flexibility and predictive performance, reflected by an effective DoF of 38.35, a log-likelihood of −252.61, and an AIC of 583.93.

GAM-derived plots (Figure 4) illustrate non-linear relationships between continuous predictors and mRS-shift. 72-h NIHSS scores and pre-mRS score were consistently associated with greater mRS-shift, aligning with trends observed in the broader cohort.

#### 3.6.3. Mortality

Within the anticoagulated cohort, logistic regression analysis (Table 7) identified NIHSS scores at admission (*p* = 0.028) and at 72 h (*p* = 0.001) as the strongest independent predictors of mortality. Neither the treatment modality nor the type of anticoagulation showed a significant association with increased mortality risk. The model demonstrated a moderate fit, with a Pseudo R^2^ of 0.3106, an improved log-likelihood of −46.014 (compared to the LL-Null of −66.743), and a highly significant overall model effect (LLR *p* = 1.5 × 10^−4^).

Regarding discriminatory performance, the model achieved an AUC of 0.87 and an overall accuracy of 84%. Specificity was high at 95%, indicating strong capability in identifying patients at lower risk of mortality; however, sensitivity was lower at 46%, reflecting a limitation in detecting all high-risk individuals. The model’s precision (0.72) and F1 score (0.57) further highlight this trade-off. Calibration was robust, as evidenced by a Brier Score of 0.1176.

## 4. Discussion

### 4.1. Summary of Findings

In this propensity score-matched cohort of patients with AIS, we found no significant differences in 90-day functional independence, mRS-shift, or mortality between anticoagulated and non-anticoagulated patients who underwent reperfusion therapy. After adjusting for confounders, including stroke severity and pre-stroke functional status, anticoagulation status was not independently associated with adverse outcomes. Furthermore, neither the type of anticoagulant nor the modality of reperfusion therapy significantly influenced clinical outcomes.

### 4.2. Interpretation and Clinical Implications

These findings challenge the widely held assumption that anticoagulated patients are inherently at greater risk for poor outcomes following reperfusion therapy. While unadjusted analyses suggested increased mortality and disability in the anticoagulated group, these differences disappeared after accounting for baseline differences.

Despite clear recommendations from major guidelines (AHA/ASA, ESO), many eligible anticoagulated patients remain untreated with reperfusion therapy in routine clinical practice. One of the reasons for this ongoing uncertainty is the limited quality of evidence supporting the safety and efficacy of reperfusion therapy in this population. Landmark reperfusion trials, such as NINDS [15] and ECASS III [2] for IVT and MR CLEAN [16], SWIFT PRIME [17], ESCAPE [18], DAWN [19], DEFUSE 3 [20] for MT either excluded anticoagulated patients or were conducted before the widespread adoption of DOACs. As a result, current guideline recommendations rely more on expert consensus than on high-quality randomized trial data [7,8,21].

### 4.3. Comparison with Existing Literature

Our results are consistent with several retrospective studies reporting comparable outcomes and ICH rates between anticoagulated and non-anticoagulated patients, including those treated outside strict guideline thresholds [22,23,24,25,26]. Interestingly, a recent meta-analysis [27] even suggested that DOAC-treated patients may experience lower rates of ICH than their non-anticoagulated counterparts, further challenging the notion that anticoagulation alone confers elevated bleeding risk. Similarly, observational studies have suggested that MT can be safely performed in anticoagulated patients [28,29,30].

Although these data are encouraging, the evidence base remains largely observational and insufficient to define definitive clinical standards. Several randomized controlled trials (RCTs) are currently underway to address this gap, including the DOAC Intravenous Thrombolysis (DO-IT) study (NCT06571149| 22 August 2024), the Safe IVT FXa (SIFT) study (NCT06878066|10 March 2025), and the ACT-GLOBAL Adaptive Platform Trial (NCT06352632|2 April 2024). While their findings are awaited, results are not anticipated in the near term. Until then, real-world guideline-conforming studies such as ours provide critical insight to support evidence-based decision-making in anticoagulated stroke patients.

### 4.4. Limitations

This study has several limitations. First, although rigorous propensity score matching was used, the retrospective observational design inherently limits causal inference and may be subject to residual confounding from unmeasured variables. Second, anticoagulation status was primarily determined through medication records and patient self-report, as point-of-care coagulation testing was not consistently available. While some patients underwent coagulation assays, DOAC plasma concentrations were not measured, and no standardized laboratory confirmation of anticoagulant activity was performed. This reliance on clinical documentation and patient recall may have introduced misclassification or recall bias, particularly in assessing treatment eligibility.

## 5. Conclusions

In summary, anticoagulation status was not independently associated with worse functional outcomes in AIS patients treated with reperfusion therapy in this observational, propensity score-matched cohort. Clinical outcomes were primarily influenced by stroke severity and pre-stroke functional status. While these findings support considering reperfusion therapy in otherwise eligible patients with prior anticoagulation, they should be regarded as hypothesis-generating and do not permit definitive conclusions about the safety and efficacy of reperfusion therapy in all anticoagulated patients. Ongoing RCTs are expected to provide more robust evidence in this population and underscore the importance of individualized, evidence-based decision-making in acute stroke care.

## Figures and Tables

**Figure 1 jcm-14-08146-f001:**
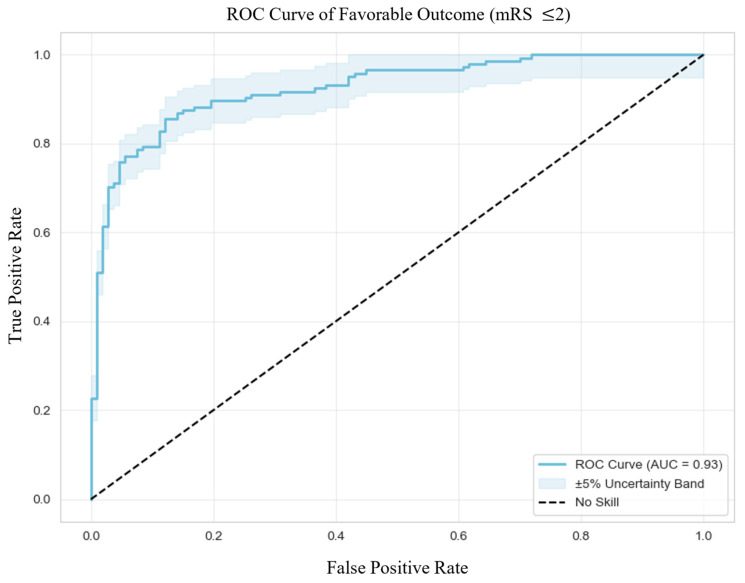
Receiver Operating Characteristic Curve of Favorable Outcome. Abbreviations: ROC = receiver operating characteristic, mRS = modified Rankin Scale, AUC = area under the curve.

**Figure 2 jcm-14-08146-f002:**
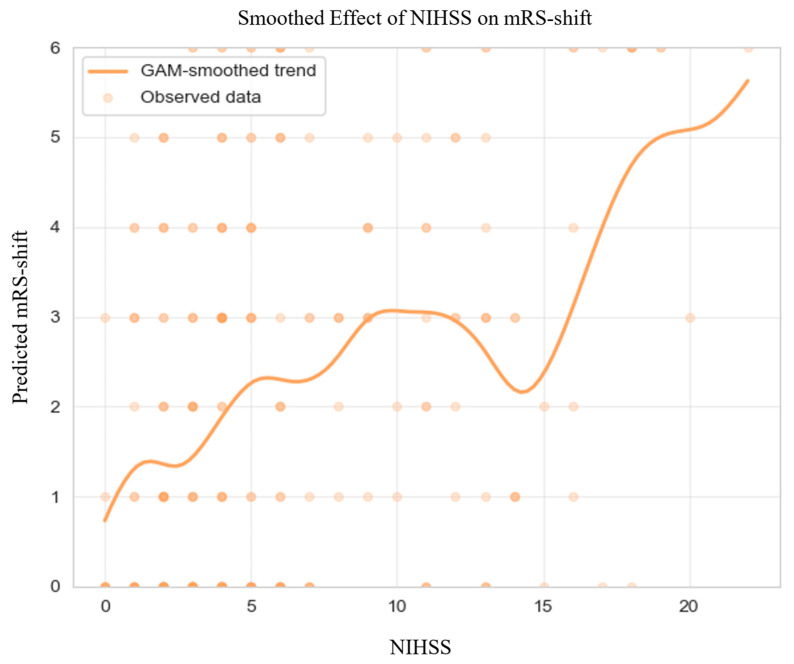
Generalized Additive Model of mRS-shift in the Matched Cohort. Abbreviations: NIHSS = National Institutes of Health Stroke Scale, mRS = modified Rankin Scale, GAM = generalized additive model. Note: Predicted mRS-shift values were truncated to the valid range of the scale (0–6) for visualization; this graphical clipping does not affect the underlying model estimates or their interpretation. The dark dots simply indicate areas with a higher density of overlapping observations.

**Figure 3 jcm-14-08146-f003:**
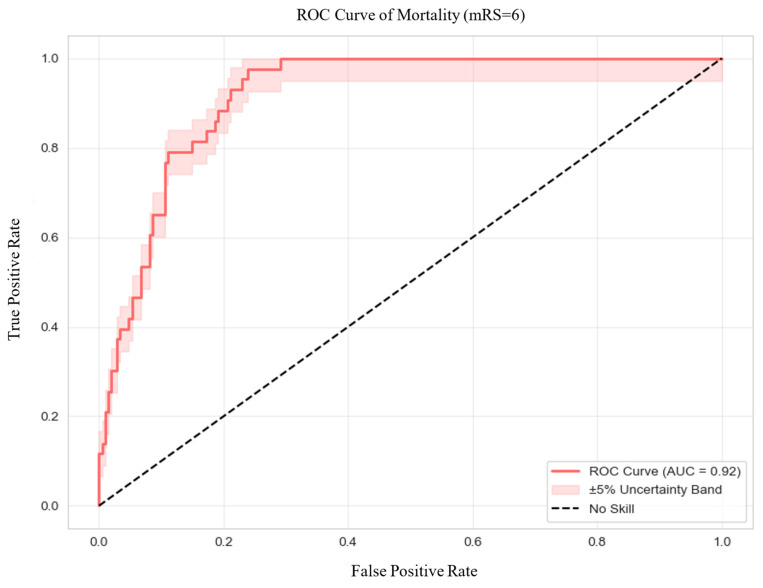
Receiver Operating Characteristic Curve of Mortality. Abbreviations: ROC = receiver operating characteristic, AUC = area under the curve.

**Figure 4 jcm-14-08146-f004:**
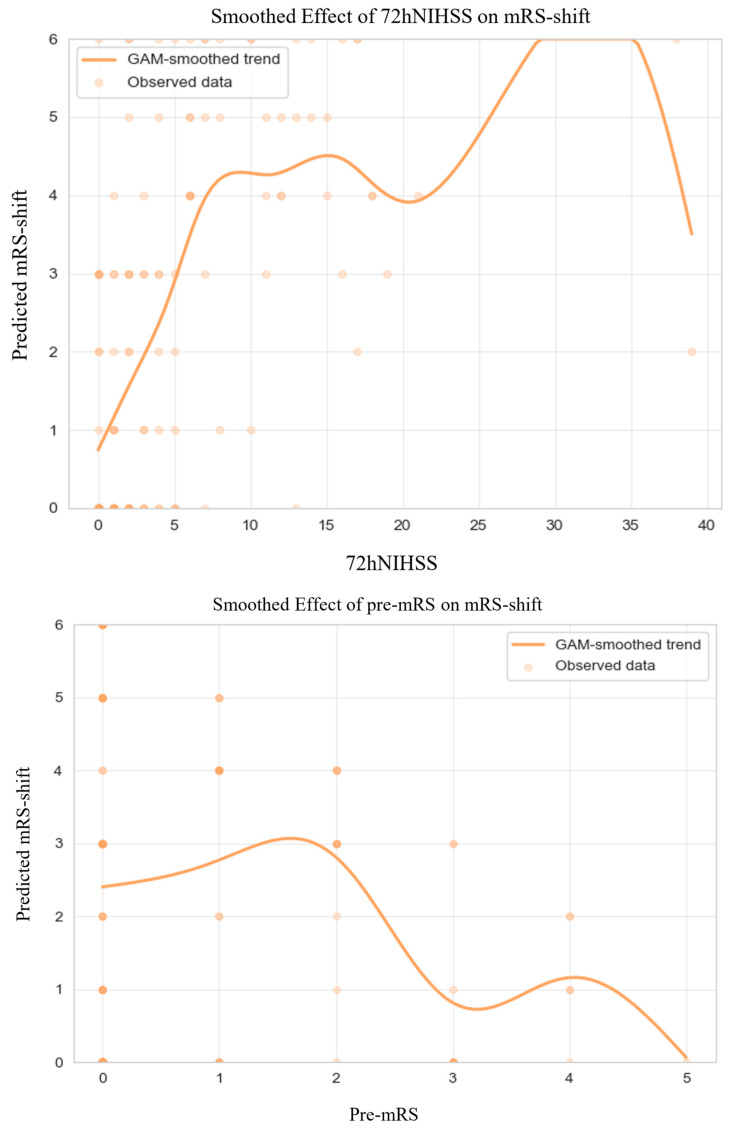
Generalized Additive Model of mRS-shift in Anticoagulated Patients. Abbreviations: NIHSS = National Institutes of Health Stroke Scale, mRS = modified Rankin Scale, GAM = generalized additive model. Note: Predicted mRS-shift values were truncated to the valid range of the scale (0–6) for visualization; this graphical clipping does not affect the underlying model estimates or their interpretation. The dark dots simply indicate areas with a higher density of overlapping observations.

**Table 1 jcm-14-08146-t001:** Baseline and Clinical Characteristics Before and After Propensity Score Matching.

	Non-Anticoagulated (*n* = 718)	Anticoagulated (*n* = 148)	*p*-Value	Matched Non-Anticoagulated (*n* = 126)	Matched Anticoagulated (*n* = 126)	*p*-Value
Demographics						
Age (years), mean ± SD	69.69 ± 12.14	76.46 ± 11.05	<0.001 *	74.94 ± 10.37	75.66 ± 10.78	0.578
Sex, male, *n* (%)	356 (49.6)	64 (43.2)	0.189	51 (40.5)	51 (40.5)	0.999
Clinical Characteristics						
Pre-mRS score, median [IQR]	0 [0–1]	0 (0–2)	0.018 *	0 (0–1)	0 (0–1)	0.889
NIHSS score at admission, median [IQR]	5 [3–8]	5 [3–11]	0.071	4 (3–7)	4 (2–9)	0.904
NIHSS score at 72 h, median [IQR]	2 [0–7]	4 [1–11]	0.030 *	3 (2–7)	3 (1–7)	0.926
ICH, *n* (%)	33 (4.6)	8 (5.4)	0.573	8 (6.4)	5 (4.0)	0.571
Etiology, cardioembolic, *n* (%)	193 (26.9)	109 (73.7)	<0.001 *	38 (30.2)	95 (75.4)	<0.001 *
Onset-to-door time (min), median [IQR]	310 [102–826]	309 [158–821]	0.503	265 [96–847]	285 [101–731]	0.865
Plasma glucose (mmol/l), mean ± SD	7.69 ± 2.96	7.45 ± 2.86	0.147	7.81 ± 3.42	7.15 ± 1.86	0.350
Medical History, *n* (%)						
Hypertension	580 (80.8)	139 (93.9)	<0.001 *	120 (95.2)	120 (95.2)	0.999
Diabetes mellitus	244 (34.0)	59 (39.9)	0.020 *	47 (37.3)	47 (37.3)	0.999
Recanalization Therapy, *n* (%)						
IVT	190 (26.5)	5 (3.4)	<0.001 *	39 (30.9)	5 (4.0)	<0.001 *
MT	124 (17.3)	39 (26.4)	0.014 *	16 (12.7)	32 (25.4)	0.016 *
IVT + MT	62 (8.6)	6 (4.1)	0.086	10 (7.9)	5 (4.0)	0.287

Abbreviations: SD = standard deviation, mRS = modified Rankin Scale, NIHSS = National Institutes of Health Stroke Scale, ICH = intracranial hemorrhage, IQR = interquartile range, IVT = intravenous thrombolysis, MT = mechanical thrombectomy. Note: * indicates statistical significance at the 0.05 level.

**Table 2 jcm-14-08146-t002:** Standardized Mean Differences (SMDs) Before and After Matching.

	Before Matching	After Matching
Age	0.583	0.067
Sex	0.127	0.000
Pre-mRS score	0.190	0.013
NIHSS score at admission	0.241	0.053
NIHSS score at 72 h	0.174	0.029
Hypertension	0.403	0.000
Diabetes mellitus	0.122	0.000

Abbreviations: SMD = standardized mean difference, mRS = modified Rankin Scale, NIHSS = National Institutes of Health Stroke Scale.

**Table 3 jcm-14-08146-t003:** Variance Ratios (VRs) and Kolmogorov–Smirnov Test Before and After Matching.

	Before Matching	After Matching
Age	0.829, *p* < 0.001 *	1.080, *p* = 0.999
Sex	0.987, *p* = 0.680	1.000, *p* = 0.999
Pre-mRS score	1.227, *p* = 0.214	0.995, *p* = 0.999
NIHSS score at admission	1.703, *p* = 0.037 *	1.294, *p* = 0.907
NIHSS score at 72 h	1.209, *p* = 0.084	1.052, *p* = 0.963
Hypertension	0.369, *p* = 0.026 *	1.000, *p* = 0.999
Diabetes mellitus	1.074, *p* = 0.764	1.000, *p* = 0.999

Abbreviations: mRS = modified Rankin Scale, NIHSS = National Institutes of Health Stroke Scale. Note: * indicates statistical significance at the 0.05 level.

**Table 4 jcm-14-08146-t004:** Multivariate Regression of Favorable Outcome in the Matched Cohort (*n* = 252).

Variable	Coefficient	*p*-Value	95% CI
Anticoagulation status	0.4417	0.346	−0.477 to 1.360
Age	0.0183	0.393	−0.024 to 0.060
Sex	0.8660	0.043 *	0.026 to 1.706
Pre-stroke mRS score	1.4207	<0.001 *	0.896 to 1.945
NIHSS score at admission	0.1072	0.035 *	0.008 to 0.207
NIHSS score at 72 h	0.5210	<0.001 *	0.343 to 0.699
Etiology, cardioembolic	−0.0044	0.992	−0.872 to 0.863
Hypertension	1.7742	0.069	−0.140 to 3.688
Diabetes mellitus	0.4666	0.256	−0.338 to 1.272
IVT	0.0903	0.865	−0.954 to 1.135
MT	0.0407	0.946	−1.128 to 1.209
IVT + MT	−0.7414	0.425	−2.564 to 1.082

Abbreviations: CI = confidence interval, mRS = modified Rankin Scale, NIHSS = National Institutes of Health Stroke Scale, IVT = thrombolysis, MT = mechanical thrombectomy. Note: * indicates statistical significance at the 0.05 level.

**Table 5 jcm-14-08146-t005:** Multivariate Regression of Mortality in the Matched Cohort (*n* = 252).

Variable	Coefficient	*p*-Value	95% CI
Anticoagulation status	0.8562	0.125	−0.237 to 1.949
Age	0.0445	0.131	−0.013 to 0.102
Sex	−0.4154	0.409	−1.401 to 0.570
Pre-stroke mRS score	−0.0785	0.673	−0.443 to 0.286
NIHSS score at admission	0.1772	<0.001 *	0.092 to 0.262
NIHSS score at 72 h	0.2075	<0.001 *	0.120 to 0.295
Etiology, cardioembolic	0.3235	0.548	−0.732 to 1.379
Hypertension	−0.1735	0.879	−2.414 to 2.067
Diabetes mellitus	0.9030	0.065	−0.056 to 1.862
IVT	0.1266	0.876	−1.469 to 1.722
MT	−0.5796	0.313	−1.706 to 0.546
IVT + MT	−0.8911	0.351	−2.764 to 0.982

Abbreviations: CI = confidence interval, mRS = modified Rankin Scale, NIHSS = National Institutes of Health Stroke Scale, IVT = thrombolysis, MT = mechanical thrombectomy. Note: * indicates statistical significance at the 0.05 level.

**Table 6 jcm-14-08146-t006:** Multivariate Regression of Favorable Outcome in Anticoagulated Patients (*n* = 126).

Variable	Coefficient	*p*-Value	95% CI
Atrial fibrillation	0.3466	0.718	−1.532 to 2.225
Previous stroke	0.5961	0.472	−1.029 to 2.222
Age	0.0026	0.946	−0.072 to 0.077
Sex	−1.0821	0.154	−2.570 to 0.406
Pre-stroke mRS score	−2.0189	0.001 *	−3.188 to −0.849
NIHSS score at admission	−0.1298	0.107	−0.288 to 0.028
NIHSS score at 72 h	−0.5987	<0.001 *	−0.915 to −0.282
Etiology, cardioembolic	−0.2166	0.784	−1.768 to 1.335
Hypertension	−2.0567	0.241	−5.491 to 1.378
Diabetes mellitus	−0.7042	0.356	−2.198 to 0.790
IVT	1.3349	0.421	−1.915 to 4.585
MT	−0.8451	0.424	−2.919 to 1.228
IVT + MT	−2.2298	0.180	−5.487 to 1.027
Type of anticoagulant	1.2983	0.140	−0.427 to 3.023

Abbreviations: CI = confidence interval, mRS = modified Rankin Scale, NIHSS = National Institutes of Health Stroke Scale, IVT = thrombolysis, MT = mechanical thrombectomy. Note: * indicates statistical significance at the 0.05 level.

**Table 7 jcm-14-08146-t007:** Multivariate Regression of Mortality in Anticoagulated Patients (*n* = 126).

Variable	Coefficient	*p*-Value	95% CI
Atrial fibrillation	0.1055	0.890	−1.386 to 1.597
Previous stroke	−0.1026	0.892	−1.586 to 1.381
Age	0.0690	0.076	−0.007 to 0.145
Sex	−1.0762	0.086	−2.306 to 0.154
Pre-stroke mRS score	−0.1025	0.690	−0.607 to 0.402
NIHSS score at admission	0.1209	0.028 *	0.013 to 0.228
NIHSS score at 72 h	0.1689	0.001 *	0.065 to 0.272
Etiology, cardioembolic	0.0117	0.987	−1.365 to 1.388
Hypertension	−0.1276	0.920	−2.603 to 2.348
Diabetes mellitus	0.8259	0.215	−0.479 to 2.130
IVT	0.5256	0.750	−2.703 to 3.754
MT	0.3485	0.623	−1.041 to 1.738
IVT + MT	1.1625	0.471	−1.998 to 4.323
Type of anticoagulant	−0.5735	0.426	−1.987 to 0.840

Abbreviations: CI = confidence interval, mRS = modified Rankin Scale, NIHSS = National Institutes of Health Stroke Scale, IVT = thrombolysis, MT = mechanical thrombectomy. Note: * indicates statistical significance at the 0.05 level.

## Data Availability

The data supporting the findings of this study are provided within the manuscript. For any additional requests regarding raw data, please contact the corresponding authors.

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
