# Peer review of "Trick or Treat(ment): Should We Still Fear Reperfusion Therapy in Anticoagulated Stroke Patients?—Comparable 90-Day Outcomes in a Propensity-Score-Matched Registry Study"

_jcm, 2025, doi:10.3390/jcm14228146_

Round 1
Reviewer 1 Report
Comments and Suggestions for Authors
Review report for “Trick or Treat(ment): Should We Still Fear Reperfusion Therapy in Anticoagulated Stroke Patients?”
First of all, the reviewer would like to thank the editors for giving me the opportunity to participate in this review process and express my respect for the authors' work, which will be important in the present field.
Comment to the Authors:
The current study investigated the risk of implementing reperfusion therapy for patients taking oral anticoagulants (DOACs) and vitamin K antagonists (VKAs) in 866 patients (DOAC n=100, VKA n=48), using a retrospective manner, using matched controls in a single center. The authors found that DOAC+VKA patients had more severe features of functional independence and mortality; however, these differences vanished after the matching. The authors further investigated that the 72-hour National Institutes of Health Stroke Scale (NIHSS) and pre-mRS were predictors of the outcome, while anticoagulation status had no significant impact.
As the authors described, the current study is limited in that it does not address the question of whether the types of stroke and the indication for reperfusion treatments are influenced by medication status. It is generally known that anticoagulated patients have a worse functional status before the onset of stroke. Additionally, the treatment targets for DOACs were not clearly defined. However, the reviewer agrees with the authors that it is important to delineate the true features of the reperfusion treatments for reassessment and further progress of acute stroke treatments.
Although the reviewer considers the current report is well written, and suited for publication in the current journal, there seem to be the following concerns that remain in the current draft;
Major concerns:
(Title) The feature of the current study should be presented in the title.
(Page 2, paragraph 2) Although the American and European guidelines recommended a significant period of time after the last anticoagulant intake, the Japanese guidelines were proposed to prioritize the clinical course and symptoms in the decision-making process for reperfusion treatments from the beginning. While the Japanese Society of Stroke is not a major player among global stroke societies, its foresight may be worth noting because the global trend followed suit.
In addition, the authors should describe the antagonists for DOACs: Idarucizumab for Dabigatran and Andexanet alfa for Rivaroxaban, Edoxaban, and Apixaban..
The application of antagonists for DOAC is not investigated.
(Tables 1 through 3) Tables 1 and 3 present nearly identical data, which are repeated and analyzed through different statistical methods. Tables 2 and 3 may be integrated.
Although the mRS shift was the core result of the current survey, the reviewer is unsure whether it is appropriate to implement such a statistical measure for variables without linearity.
Minor concerns
Page 1, line 1, “morbidity” is unclear in meaning. Please check.
Reviewer 2 Report
Comments and Suggestions for Authors
Here are some comments about this paper-
- In this paper, the reperfusion therapy showed positive effects on acute stroke care, and we are curious about whether the recanalization therapy shows the similar effects in other stroke like diseases such as traumatic brain injury, which might expand the application scope.
2. Have you ever considered the inflammation response during the reperfusion therapy? We wonder whether this therapy play positive or negative roles in inflammatory complications during stroke treatment, or whether the roles are individualized?
